

# Satellite data reveal details of glacial isostatic adjustment in the Amundsen Sea Embayment, West Antarctica

Matthias O. Willen[1,a], Bert Wouters[1], Taco Broerse[1], Eric Buchta[2], and Veit Helm[3]

[1]Department of Geoscience and Remote Sensing, Delft University of Technology, The Netherlands
[2]Institut für Planetare Geodäsie, Technische Universität Dresden, Germany
[3]Alfred Wegener Institute, Helmholtz Centre for Polar and Marine Research, Bremerhaven, Germany
[a]now at: 2

**Correspondence:** Matthias O. Willen (matthias.willen@tu-dresden.de)

**Abstract**: The instability of the West Antarctic ice sheet (WAIS) is a tipping element in the climate system and it is mainly dictated by changes in the ice flow behavior of the outflow glaciers in the Amundsen Sea Embayment (ASE). Recent studies postulated that vertical uplift of bedrock can delay the collapse of glaciers in this region. In West Antarctica, bedrock motion is largely caused by a fast viscoelastic response of the upper
mantle to changes in ice loads during the last centuries. This glacial isostatic adjustment (GIA) effect is poorly understood so far, since Earth's rheology and the ice-loading history are both subject to large uncertainties in simulations. Moreover, results from data-driven approaches have not yet resolved GIA at a sufficient spatial resolution. We present a data-driven GIA estimate, based on data from GRACE/GRACE-FO, CryoSat-2 altimetry, regional climate modelling, and firn modelling, that is the first to agree with independent GNSS-derived
vertical velocities in West Antarctica. Our data combination yields a maximum GIA bedrock-motion rate of $43 \pm 7\,\mathrm{mm\,a^{-1}}$ in the Thwaites Glacier region and agrees within uncertainties of the GNSS-derived rate. The data-driven present-day GIA result may be used in future simulation runs to quantify a potential delay of the collapse of the West Antarctic ice sheet due to the stabilization effects induced by GIA. Furthermore it may be used for testing rheological models with a low upper-mantle viscosity in conjunction with centennial loading
histories.

## 1  Introduction

The shrinking of ice sheets due to a warming climate and its contribution to sea level rise is a major public concern. The West Antarctic ice sheet (WAIS) warrants particular focus, because, if the global mean temperature exceeds 1.5°C relative to the pre-industrial time, a threshold may be be reached at which the WAIS becomes
unstable (McKay et al., 2022). The instability means that the glacier flow accelerates abruptly, leading to a major outflow of ice from the WAIS into the ocean. This 1.5°C threshold is very likely to be reached within the next 20 years even considering low-emission scenarios (IPCC, 2021). Furthermore, ice-ocean interaction



simulations demonstrated that any reduction of greenhouse gases has a very limited impact to prevent WAIS' accelerated contribution to sea-level rise during the next decades (Naughten et al., 2023).

A changing bedrock topography due to solid-Earth deformation may affect the glacier flow and thus the outflow flux (Whitehouse et al., 2019). High bedrock uplift rates of several centimetres per year in the region of
the Amundsen Sea Embayment (ASE) are evident from GNSS measurements (Groh et al., 2012) and this uplift may provide a feedback that stabilizes the WAIS in the future (Adhikari et al., 2014; Gomez et al., 2015; Konrad et al., 2015; Book et al., 2022). Namely, bedrock uplift leads to a shift of the grounding line—the boundary between the grounded ice on the continent and the floating ice—towards the ocean. If the grounding line moves towards the ocean, a larger part of the glacier ice will be grounded on the continent and the ice thickness at the
grounding line will be smaller. A smaller ice thickness results in less ice outflow, resulting in a stabilization of the ice sheet (Whitehouse et al., 2019). The GNSS observations that provide estimates for this bedrock uplift are subject to limitations: in particular, a restriction to bedrock outcrops leading to a coarse spatial resolution, and expensive logistics, with the consequence that many sites have at a low temporal sampling. Barletta et al. (2018) modelled glacial isostatic adjustment (GIA) to fit the high bedrock uplift rates observed with GNSS in
the ASE. They show that the GNSS observations can be explained by a mantle response to ice changes of the last 100 years, when adopting a very low upper mantle viscosity. Global GIA models only barely explain the observed uplift rate in the ASE, as they typically utilize a rheology and an ice-loading history that are outside the range of parameters relevant for this region (Whitehouse et al., 2019).

For the Antarctic ice sheet, there is considerable difference amongst various models of present-day GIA and
its induced mass effect (Whitehouse et al., 2019). To elucidate: Groh and Horwath (2021) estimate a gravimetric ice mass balance of the AIS of $-91 \pm 44\,\mathrm{Gt\,a^{-1}}$ from Apr 2002 until Jul 2020. The uncertainty in the present-day GIA mass effect contributes to about 3/4 of the indicated total mass-balance uncertainty (Groh and Horwath, 2021). Data combination approaches, often called inverse approaches, estimate the GIA-induced mass changes by utilizing satellite gravimetry and satellite altimetry observations. These GIA estimates are useful to improve
the ice mass change (IMC) estimates of the AIS (Willen et al., 2024). However, published inverse GIA estimates that do not incorporate GNSS data explain only a part of the bedrock uplift in the ASE. These approaches resolve GIA at a coarse spatial resolution only, at a level that is insufficient to explain the GNSS observations in the ASE (for an overview see Whitehouse et al. (2019)). For example, combination approaches from Riva et al. (2009), Gunter et al. (2014), Engels et al. (2018), Willen et al. (2024) can only resolve GIA at an effective spatial
resolution of $>400\,\mathrm{km}$, insufficient to capture the GIA effect with spatial scales of $\sim100\,\mathrm{km}$ as postulated by Barletta et al. (2018). This coarse resolution is mainly a consequence of processing choices informed by the data quality. The shortcoming to explain the GNSS-derived uplift magnitudes and small spatial scales also holds for the inverse approaches that incorporate GNSS data in addition to gravimetry and altimetry data (Martín-Español et al., 2016b; Martín-Español et al., 2016a; Sasgen et al., 2017). Furthermore, including GNSS data
directly in the inversion removes the ability to independently validate GIA estimates.




So far it was not possible to resolve GIA-related bedrock motion in the ASE in a realistic order of magnitude independently from GNSS measurements. Here, we investigate how we can quantify a realistic GIA with observations spatially continuously covering the whole ASE. Given by the high signal-to-noise ratio in the ASE region and improved data processing, we hypothesize that a data combination approach on time-series
level can achieve this. We develop this combination approach according to Willen et al. (2020), which itself builds upon the approach of Gunter et al. (2014), while we avoid unrealistic spatial scale constraints. By that, we can provide a spatially continuous description of GIA in the ASE, that supplements the sparse sampling of GNSS-based GIA estimates. To do so, we make use of 10 years of available elevation changes from CryoSat-2 data, gravitational field changes from GRACE/GRACE-FO, and regional climate modelling (RACMO2) as well
as firn modelling (IMAU-FDM) outputs. We restrict the analysis here to the elevation changes of CryoSat-2, because earlier altimetry missions have either a limited spatial sampling by orbit design (Envisat) or a limited temporal sampling by its mission concept (ICESat). In addition, CryoSat-2 offers high data quality (Schröder et al., 2017). Thereby we accept to be limited to a time span of 10 years. We use GNSS data only to validate our GIA estimate, and do not include it in the estimation procedure.

## 2  Data

Our data combination approach uses data from the GRACE and GRACE-FO satellite missions. These are monthly gravity field changes (Sect. 2.1). Next, it includes monthly grids of elevation changes derived from the radar altimetry mission CryoSat-2 (Sect. 2.2). Lastly, the approach incorporates modelling outputs from the regional climate model RACMO2 and the firn model IMAU-FDM. More precisely, the data combination uses
monthly changes in firn air content (FAC) derived from the modelling outputs (Sect. 2.3). The firn layer of an ice sheet can be conceptually divided into two components: ice and air. FAC represents the air component, expressed as an equivalent height. Sect. 3 details how FAC relates to the observations and how the combination approach includes FAC. The use of FAC over firn density benefits a data combination approach, as previously demonstrated Willen et al. (2022). GNSS data serve to validate the results (Sect. 2.4). All data sets are available
with at least monthly temporal resolution. With regard to GIA-related deformation, such a high temporal resolution is presumably not necessary. However, we combine the data sets at a monthly temporal resolution, as we do not aim here to implement any a priori assumptions about the temporal behavior of the signals. All of the following subsections describe how we use monthly uncertainty information from the data sets. In all cases where we provide rate estimates, the corresponding rate uncertainty follows from a full error-covariance
propagation using the analogous data combination on trend level. We use the error covariances of all input datasets from Willen et al. (2022) to estimate rate uncertainties.

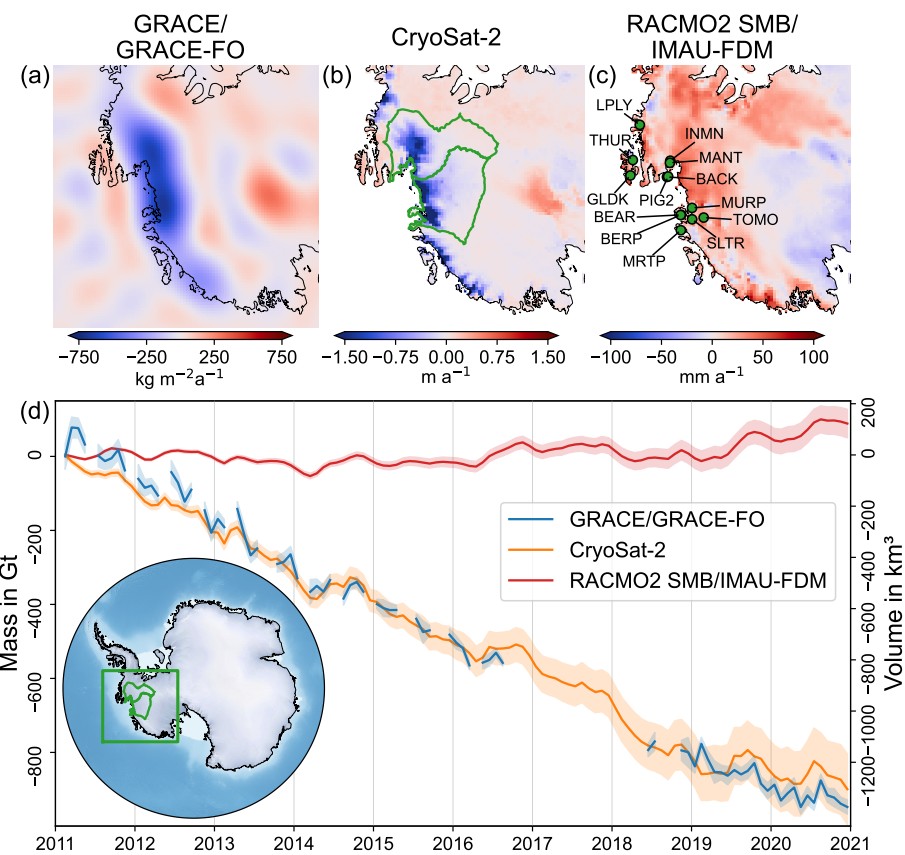

**Figure 1:** The original input data in the study region. Mean rates from Jan 2011 to Dec 2020 of (a) surface density rates from GRACE/GRACE-FO data (Mayer-Gürr et al., 2018), (b) of surface elevation rates from CryoSat-2 (Helm et al., 2014; Helm et al., 2024), and (c) rates of the firn air content (FAC) from IMAU-FDMv1.2A (Veldhuijsen et al., 2023) and RACMO2.3p2 SMB (Wessem et al., 2018) (c). GNSS-site locations where data is used for validation purposes are illustrated with green circles and labelled with site names in (c), (see Table 1). The drainage area of the Amundsen Sea Embayment (ASE), i.e. basin 21 and 22 according to Zwally et al. (2012) is indicated with a green polygon in (b). (d) Time series of integrated observations over the ASE region including an overview of the investigation area. Left y-axis mass change (GRACE/GRACE-FO), right y-axis volume change (CryoSat-2 and RACMO2 SMB/IMAU-FDM).



## 2.1 Satellite gravimetry

We utilize *ITSG-Grace2018* and *ITSG-Grace_op* monthly gravity field solutions up to degree and order 96 (Mayer-Gürr et al., 2018). Ditmar (2022) found that these solutions outperform other gravity field solutions in terms of noise level and signal retainment. Spherical harmonic coefficients of degree one complement the gravitational field, following the approach from Sun et al. (2016). We replace i) the spherical harmonic coefficient of degree two and order zero with Satellite Laser Ranging products (Loomis et al., 2020) for all gravitational fields and ii) the coefficient of degree three and order zero for gravitational fields obtained during GRACE/GRACE-FO accelerometer failures. We express these gravitational field changes as surface density changes (Eq. 3) on the WGS84 ellipsoidal surface, applying the approach from Ditmar (2018), and transfer them to the spatial domain on a 20 km x 20 km polarstereographic grid in West Antarctica (Figure 1a).

We propagate the full error-covariance information (Koch, 1999) provided along with the ITSG monthly gravity field solutions (Mayer-Gürr et al., 2018) to the spatial domain, i.e. to the surface density changes using the ellipsoidal surface approximation. Gaussian smoothing suppresses the typical spatially correlated GRACE/GRACE-FO error patterns. A full propagation of the spatially correlated errors is certainly the most complete approach, yet, it leads to very high computational efforts. Therefore, on time series level of the data combination, we simplify the error propagation and restrict ourselves to the spatially uncorrelated error parts. Our error estimates thus serve as an upper bound.

## 2.2 Satellite altimetry

We obtain monthly grids of surface elevation changes from CryoSat-2 data processed using an updated approach according to Helm et al. (2014), i.e. the elevation change is derived using a TFMRA retracker and corrected with sigma correlation (cf. Sect. 2.4 in Helm et al. (2024)). We resample the surface elevation changes to the same 20 km x 20 km polarstereographic grid in West Antarctica as we use for evaluation of GRACE/GRACE-FO data in the spatial domain.

As there is no official uncertainty product, we assess the uncertainty of surface elevation time series by comparing data from CryoSat-2 and ICESat-2 during the four-year period from 2019-01 to 2022-12, when both altimetry missions were observing simultaneously. We assume two types of uncertainties: (1) temporally uncorrelated, i.e. white noise, and (2) temporally correlated over the full observation period, i.e. a trend uncertainty. The uncorrelated uncertainty (1) is quantified by the standard deviation of the residuals that result by fitting a deterministic trend-cycle model (bias, linear, quadratic, annual cycles, semi-annual cycles) to CryoSat-2 and ICESat-2 differences. We quantify the trend uncertainty (2) by calculating the rate and the acceleration of differences between CryoSat-2 and ICESat-2. This trend uncertainty is relative to January 2011, i.e. at this time, the trend uncertainty is zero and increases over time.



## 2.3 Firn Air Content (FAC) changes

To obtain the time series of the FAC change (e.g. Ligtenberg et al., 2014), we use an updated version of the RACMO2.3p2 SMB output (Wessem et al., 2018, retrieved at 2021-11-30) and firn-thickness changes from the IMAU-FDM v1.2A (Veldhuijsen et al., 2023). We assume that the mass change of the firn layer equals the SMB change. Theoretically there could be also a SMB-driven mass change of the ice layer which we neglect here. From the firn-thickness change and SMB, we calculate the change in FAC (Eq. 6). Similar as the previous datasets, FAC changes are resampled to the same 20 km x 20 km polarstereographic grid.

We characterize the uncertainty of the FAC time series using an alternative FAC product from Goddard Space Flight Center (GSFC) the GSFC-FDM (Medley et al., 2022) for comparison. Here, we also assume two types of uncertainties: uncorrelated in time and fully correlated in time over the observation period. For quantification, we follow the same approach as we apply to surface elevation changes from satellite altimetry (Sect. 2.2). To do so, we evaluate differences in FAC changes from the IMAU-FDM and the GSFC-FDM.

## 2.4 GNSS data

The GNSS data originates from sites in Antarctica with GNSS antennas mounted on bedrock outcrops, directly observing horizontal and vertical bedrock displacement. The data combination method (Sect. 3) only allows the estimation of vertical bedrock motion, so that we do not consider horizontal displacements further. There are two differently conceptualised GNSS setups to monitor bedrock motion. These include (1) continuous sites (*cont* in Table 1), which are designed to enable continuous measurements over several years. On the other hand, there are (2) episodic sites (*epis* in Table 1), at which recurring campaign measurements are realized at fixed anchored locations. The campaigns are repeated after several years, with data usually being collected over several days in each individual campaign (Scheinert et al., 2021). With (1) the motion of the surface of the solid Earth can be studied at a high temporal resolution. The campaign-style experiments (2) aim by design to provide long-term rates of bedrock motion only. At some sites were initally designed as episodic setups and later upgraded to continuously operating sites. (*both* in Table 1).

We include 13 GNSS sites in this study. The sites are either affiliated to the POLENET-ANET network (Wilson and Bevis et al., 2019) or result from measurement campaigns conducted by TU Dresden. The GNSS-derived bedrock motions result from a consistent Antarctic-wide analysis that has been accomplished in the frame of the SCAR-endorsed Geodynamics In ANTarctica based on REprocessing GNSS dAta INitiative (GIANT-REGAIN, Buchta et al., 2024). Here we use the processing results of the individual TU Dresden solution. We refer the reader to Buchta et al. (2024) for all details of GNSS data processing. For each month, $m$, we calculate the weighted mean of all daily solutions in that month:



**Table 1:** Overview of used GNSS data in the Amundsen Sea Embayment that was analyzed by Buchta et al. (2024). The first four columns provide the GNSS site ID, their coordinates (Lon and Lat) as well as the type of GNSS observations (continuous (cont) site or a episodic (epis) site or a mixture of both (both)). The fifth column gives the number of campaigns (noc). The columns (start) and (end) list when the first and last observations were taken. Column (t dur.) specifies the total time span rounded to full years. Vertical bedrock motion rates, and its split into the elastic-related and GIA-related component (Eq. 11) are provided in columns ($\dot{h}^{\mathrm{BM}}$), ($\dot{h}^{\mathrm{ELA}}$) and ($\dot{h}^{\mathrm{GIA}}$), respectively.

| GNSS site | Lon | Lat | Type | noc | start | end | t dur. | $\dot{h}^{\mathrm{BM}}$ | $\dot{h}^{\mathrm{ELA}}$ | $\dot{h}^{\mathrm{GIA}}$ |
| --- | --- | --- | --- | --- | --- | --- | --- | --- | --- | --- |
| | in ° | in ° | | | | | in a | in mm a$^{-1}$ | in mm a$^{-1}$ | in mm a$^{-1}$ |
| BACK | $-102.478$ | $-74.430$ | both | 1 | 2006-01 | 2020-12 | 15 | $15.9 \pm 1.2$ | $4.98 \pm 0.03$ | $11.0 \pm 1.2$ |
| BEAR | $-111.888$ | $-74.579$ | epis | 2 | 2006-03 | 2010-03 | 4 | $24.9 \pm 2.1$ | $4.81 \pm 0.06$ | $20.0 \pm 2.1$ |
| BERP | $-111.885$ | $-74.546$ | both | 1 | 2003-11 | 2020-12 | 17 | $26.2 \pm 2.1$ | $6.06 \pm 0.05$ | $20.1 \pm 2.1$ |
| GLDK | $-100.588$ | $-72.233$ | cont | | 2018-12 | 2020-12 | 2 | $-5.3 \pm 1.6$ | $-0.99 \pm 0.05$ | $-4.3 \pm 1.6$ |
| INMN | $-98.880$ | $-74.821$ | cont | | 2013-01 | 2019-12 | 7 | $32.3 \pm 2.6$ | $8.69 \pm 0.19$ | $23.6 \pm 2.6$ |
| LPLY | $-90.299$ | $-73.111$ | both | 1 | 2006-01 | 2020-12 | 15 | $5.8 \pm 0.6$ | $2.93 \pm 0.02$ | $2.9 \pm 0.6$ |
| MANT | $-99.368$ | $-74.779$ | epis | 2 | 2006-02 | 2017-02 | 11 | $29.0 \pm 2.1$ | $8.42 \pm 0.17$ | $20.6 \pm 2.1$ |
| MRTP | $-115.102$ | $-74.180$ | cont | | 2018-12 | 2020-12 | 2 | $15.1 \pm 2.0$ | $2.62 \pm 0.03$ | $12.5 \pm 2.0$ |
| MURP | $-111.294$ | $-75.369$ | epis | 2 | 2006-03 | 2016-01 | 10 | $62.9 \pm 4.7$ | $15.88 \pm 0.49$ | $47.0 \pm 4.7$ |
| PIG2 | $-102.439$ | $-74.511$ | epis | 2 | 2006-03 | 2017-02 | 11 | $16.8 \pm 1.2$ | $5.11 \pm 0.03$ | $11.7 \pm 1.2$ |
| SLTR | $-113.880$ | $-75.098$ | cont | | 2018-12 | 2020-12 | 2 | $51.1 \pm 4.5$ | $11.18 \pm 0.69$ | $39.9 \pm 4.6$ |
| THUR | $-97.560$ | $-72.530$ | both | 1 | 2006-01 | 2020-12 | 15 | $-2.2 \pm 0.6$ | $1.73 \pm 0.03$ | $-4.0 \pm 0.6$ |
| TOMO | $-114.662$ | $-75.802$ | cont | | 2012-01 | 2020-12 | 9 | $52.0 \pm 3.9$ | $13.54 \pm 0.41$ | $38.5 \pm 4.0$ |

$$h_m^{\mathrm{GNSS}} = \frac{\sum_{d=1}^{D} h_d^{\mathrm{GNSS}}(\sigma_d^{\mathrm{GNSS}})^{-2}}{\sum_{d=1}^{D} (\sigma_d^{\mathrm{GNSS}})^{-2}} \tag{1}$$

$h_d^{\mathrm{GNSS}}$ and $\sigma_d^{\mathrm{GNSS}}$ refer to the GNSS-derived vertical component available at a day, $d$, and its uncertainty, respectively. $D$ is the number of available daily solutions in a certain month, $m$.

To derive a representative monthly uncertainty, $\sigma_m^{\mathrm{GNSS}}$, we use the standard uncertainty of the weighted mean, which is derived from the uncertainties of the individual daily solutions. However, the uncertainty of each daily solution is limited to represent all error effects (Buchta et al., 2024). For this reason, we add the variance of all daily solutions in this month:

$$(\sigma_m^{\mathrm{GNSS}})^2 = \left[ \sum_{d=1}^{D} (\sigma_d^{\mathrm{GNSS}})^{-2} \right]^{-1} + \frac{1}{D-1} \sum_{d=1}^{D} (h_d^{\mathrm{GNSS}} - h_m^{\mathrm{GNSS}})^2 \,. \tag{2}$$

## 3   Regional data combination method

In order to quantify GIA effects in the ASE, we apply a data combination method similar to the method presented by Gunter et al. (2014) and that was extended to a combination on time-series level by Willen et al. (2020). We utilize observations of satellite gravimetry and satellite altimetry as well as results of regional climate modelling and firn modelling (Sect. 2). Further we use GNSS data for validation. Signals to be separated




are from the processes GIA and IMC. The data combination method builds upon the different sensitivity of the data sets towards the signals to be separated. This sensitivity is given by the effective densities between the physical quantities that change due to the processes explained below. We aim to solve for the physical quantity of bedrock motion that contemporary changes due to GIA ($\boldsymbol{h}^{\mathrm{GIA}}$) and we co-estimate the surface density change

due to IMC ($\boldsymbol{\kappa}^{\mathrm{IMC}}$).

## Surface density changes

A time series of monthly surface density changes, $\boldsymbol{\kappa}$ with the unit $[\kappa] = \mathrm{kg\,m^{-2}a^{-1}}$, loosely also referred to as mass changes, derived from satellite gravimetry, $\boldsymbol{\kappa}^{\mathrm{GRAV}}$, contain the following quantities:

$$\boldsymbol{\kappa}^{\mathrm{GRAV}} = \boldsymbol{\kappa}^{\mathrm{GIA}} + \boldsymbol{\kappa}^{\mathrm{IMC}} + \boldsymbol{\kappa}^{\mathrm{OTHER}} + \boldsymbol{\varepsilon}^{\mathrm{GRAV}}, \tag{3}$$

$\boldsymbol{\kappa}^{\mathrm{GRAV}}$ originate from monthly gravitational fields provided as Stokes coefficients and are evaluated on an
ellipsoidal surface to retrieve surface density changes (Ditmar, 2018). The potential change related to elastic deformation is accounted for when converting potential changes to surface density changes. $\boldsymbol{\kappa}^{\mathrm{GIA}}$ and $\boldsymbol{\kappa}^{\mathrm{IMC}}$ are surface density changes related to GIA and IMC, respectively. $\boldsymbol{\kappa}^{\mathrm{OTHER}}$ refers to far-field effects from mass changes from all other regions on the Earth that result from the evaluation of gravitational field changes (Willen et al., 2024). In the ASE, we assume that these far-field effects are very small compared to the mass variations
taking place in the ASE and can be neglected. $\boldsymbol{\varepsilon}^{\mathrm{GRAV}}$ refers to the observational error of satellite gravimetry.

Ice mass change (IMC), $\boldsymbol{\kappa}^{\mathrm{IMC}}$, is the sum of mass changes in the firn layer, $\boldsymbol{\kappa}^{\mathrm{FIRN}}$, and in the ice layer, $\boldsymbol{\kappa}^{\mathrm{ICE}}$. Mass changes in the firn layer are explained by the surface mass balance (SMB), e.g. modelled with a regional climate model (Sect. 2). We assume that $\boldsymbol{\kappa}^{\mathrm{FIRN}} \sim \boldsymbol{\kappa}^{\mathrm{SMB}}$:

$$\boldsymbol{\kappa}^{\mathrm{IMC}} = \boldsymbol{\kappa}^{\mathrm{SMB}} + \boldsymbol{\kappa}^{\mathrm{ICE}} \tag{4}$$

## Surface elevation changes

Time series of surface elevation changes, $\boldsymbol{h}$ with the unit $[h] = \mathrm{m\,a^{-1}}$, observed with satellite altimetry, $\boldsymbol{h}^{\mathrm{ALT}}$, contain the following signals:

$$\boldsymbol{h}^{\mathrm{ALT}} = \boldsymbol{h}^{\mathrm{IMC}} + \boldsymbol{h}^{\mathrm{FAC}} + \boldsymbol{h}^{\mathrm{GIA}} + \boldsymbol{h}^{\mathrm{ELA}} + \boldsymbol{\varepsilon}^{\mathrm{ALT}}, \tag{5}$$

$\boldsymbol{h}^{\mathrm{GIA}}$ and $\boldsymbol{h}^{\mathrm{ELA}}$ refer to GIA-induced and elastic-deformation-induced bedrock motion, respectively. $\boldsymbol{h}^{\mathrm{IMC}}$ is the surface elevation change due to IMC. $\boldsymbol{h}^{\mathrm{FAC}}$ refers to the change of firn air content (FAC):





$$h^{\text{FAC}} = h^{\text{FIRN}} - \frac{\kappa^{\text{SMB}}}{\rho^{\text{ICE}}} \tag{6}$$

We obtain $h^{\text{FAC}}$ from modelled firn thickness variations (Sect. 2), $h^{\text{FIRN}}$, and express the modelled cumulated SMB anomalies as a purely solid-ice-related elevation change. $\rho^{\text{ICE}}$ is the density of pure ice and is assumed to be $917\,\text{kg m}^{-3}$. Thereby we assume that sum of elevation changes due to IMC and FAC ($h^{\text{IMC}} + h^{\text{FAC}}$) equals the sum of surface elevation changes caused by changes in ice-flow dynamics and firn thickness ($h^{\text{IFD}} + h^{\text{FIRN}}$).

## 5 Combining surface elevation and surface density changes

The ratio of the surface density change and the surface elevation change caused by GIA has the unit of a density, and is referred to as the effective GIA density, $\rho^{\text{GIA}}$:

$$\kappa^{\text{GIA}} = \rho^{\text{GIA}} h^{\text{GIA}}, \tag{7}$$

We use a spatial mask given the GIA at each location in Antarctica according to Riva et al. (2009), which is based on findings from GIA modelling (Fig. S2). Using Eq. 7 and the relation:

$$\kappa^{\text{IMC}} = \rho^{\text{ICE}} h^{\text{IMC}}, \tag{8}$$

, while leaving out the error components, we deterministically combine surface density changes and elevation changes as follows to separate GIA-related surface density changes and surface elevation changes, respectively:

$$h^{\text{GIA}} = \left( h^{\text{ALT}} - h^{\text{FAC}} - h^{\text{ELA}} - \frac{\kappa^{\text{GRAV}}}{\rho^{\text{ICE}}} \right) \frac{\rho^{\text{ICE}}}{\rho^{\text{ICE}} - \rho^{\text{GIA}}} \tag{9}$$

$$\kappa^{\text{GIA}} = \rho^{\text{GIA}} \cdot h^{\text{GIA}}. \tag{10}$$

This assumes that $\kappa^{\text{OTHER}} = 0$ in Eq. 3 as mentioned above. We approximate $h^{\text{ALT}} - h^{\text{FAC}} - h^{\text{ELA}}$ with $1.015\,(h^{\text{ALT}} - h^{\text{FAC}})$ (Riva et al., 2009).

The GIA-related mean rate of surface density changes and surface elevation changes, $\dot{\kappa}^{\text{GIA}}$ and $\dot{h}^{\text{GIA}}$, respectively, can be obtained from $\kappa^{\text{GIA}}$ and $h^{\text{GIA}}$ via least-squares adjustment of a trend-seasonal model. Co-estimated seasonal (annual and semi-annual) components capture potential errors that have propagated to the GIA result.

The error covariance information from Willen et al. (2024) for the mean rates of the input datasets provides





a more realistic error information than what is available for data on the time-series level. To estimate more realistic uncertainties of the estimated GIA-related mean rates, we adapt the time-series combination approach (Eq. 9) to a trend-level combination approach as in Gunter et al. (2014). To do so, one needs to first determine the mean rates from a least-squares adjustment of the input data sets ($\dot{h}^{\mathrm{ALT}}$, $\dot{h}^{\mathrm{FAC}}$, $\dot{\kappa}^{\mathrm{GRAV}}$) and then combine these

analogically as shown in Eq. 9 to estimate $\dot{\kappa}^{\mathrm{GIA}}$ and $\dot{h}^{\mathrm{GIA}}$. Note that we use the combination on trend-level for propagating error covariances only.

## Optimal spatial unification of input data

To unify the spatial resolution of the input data (Gunter et al., 2014), we first apply a Gaussian smoother to each data set. This step is most likely legitimate for determining GIA effects, as these occur at longer

spatial wavelengths than the resolution of satellite altimetry ($<10\,\mathrm{km}$). According to modelling results from Barletta et al. (2018), even GIA associated with a low viscosity in the upper-mantle and centennial ice loading changes occurs on spatial wavelengths larger than $100\,\mathrm{km}$ (cf. Fig. S13 in Barletta et al. (2018)). However, IMC takes place on much smaller spatial scales. By combining previously smoothed data, we can only determine a spatially smoothed $\kappa^{\mathrm{IMC}}$.

It is initially unknown, what Gaussian filter width (here referred to as the half-response width) is optimal for separating GIA from IMC such that the spatial resolution is close to the true GIA effect. If the filter width is too large, the true GIA signal may be overly smoothed. A filter width that is too small may lead to an insufficient unification of the spatial resolution, so that the result is dominated by artefacts and spatial noise. To identify what filter width is optimal, we compare the GIA result from the combination (Eq. 9) with independent GNSS

data. GNSS observations $h^{\mathrm{GNSS}}$ observe the full bedrock motion, $h^{\mathrm{BM}}$, which contains both GIA and elastic contributions:

$$h^{\mathrm{GNSS}} = h^{\mathrm{BM}} + \varepsilon^{\mathrm{GNSS}} = h^{\mathrm{GIA}} + h^{\mathrm{ELA}} + \varepsilon^{\mathrm{GNSS}}. \tag{11}$$

## Benchmarking of the GIA estimate against GNSS data

Although GNSS data provide information at single observation sites only, they provide a full pointwise measurement of the bedrock motion magnitude at this position. This makes them ideal for validating the combina-

tion results (e.g. Kappelsberger et al., 2024). However, vertical bedrock motion from GNSS contains not only GIA effects but also elastic deformation effects (Eq. 11). These elastic effects take place on smaller spatial scales than GIA (Farrell, 1972). The smoothing of the data sets we perform is useful for detecting GIA signals, but not for resolving elastic deformation effects at a spatial resolution high enough to be comparable to GNSS data. For this reason, we determine high-resolution elastic deformation effects from the unsmoothed altimetry





observations. To do so, we approximate high-resolution IMC as follows:

$$\boldsymbol{h}^{\mathrm{IMC}} = 1.015(\boldsymbol{h}^{\mathrm{ALT}} - \boldsymbol{h}^{\mathrm{FAC}}) - \boldsymbol{h}^{\mathrm{GIA}}, \tag{12}$$

using unsmoothed $\boldsymbol{h}^{\mathrm{ALT}}$, unsmoothed $\boldsymbol{h}^{\mathrm{FAC}}$, and $\boldsymbol{h}^{\mathrm{GIA}}$ from Eq. 9. This high-resolution approximation is used to determine the elastic deformation effects by using the Green's function approach in the spatial domain (Farrell, 1972). We use tabulated Green's function computed from the Preliminary Reference Earth Model

5    (PREM, Dziewonski and Anderson, 1981; Wang et al., 2012) in the centre of figure frame. This approach is inconsistent to some degree but has a negligible impact on the result in this region (Sect. S1).

In addition to comparing the full bedrock motion from the data combination (GIA + elastic) and GNSS, we also compare GIA-only bedrock motion from the data combination (Eq. 9) to elastic-corrected GNSS data and the simulation results of Barletta et al. (2018). In contrast to the comparison of full bedrock motion, the

10   comparison of GIA from the data combination with GIA from GNSS data is not completely independent, because the GIA result and the elastic-corrected GNSS data depend on the same altimetry data. Furthermore, the model from Barletta et al. (2018) that we compare to is tailored to best explain GNSS data. It has overlap with the GNSS data that we use (Sect. 2), yet it has been processed differently.

We assess the agreement between the combination result and the GNSS data in terms of weighted root mean

15   square differences (WRMSD):

$$\mathrm{WRMSD}\left(\dot{h}^{\mathrm{BM}}\right) = \sqrt{\frac{\sum \left[ w_i \left( \dot{h}^{\mathrm{BM}}_{i,\mathrm{COMB}} - \dot{h}^{\mathrm{BM}}_{i,\mathrm{GNSS}} \right)^2 \right]}{\sum w_i}}, \tag{13}$$

with the weight, $w$, for each GNSS site, $i$:

$$w_i = \frac{1}{\sigma^2_{i,\mathrm{COMB}} + \sigma^2_{i,\mathrm{GNSS}}}. \tag{14}$$

To retrieve $\dot{h}^{\mathrm{BM}}_{i,\mathrm{COMB}}$ the combination result from Eq. 9 is evaluated at the location of each GNSS site, $i$. $\sigma^2$ refers to the variance as a measure of uncertainty.



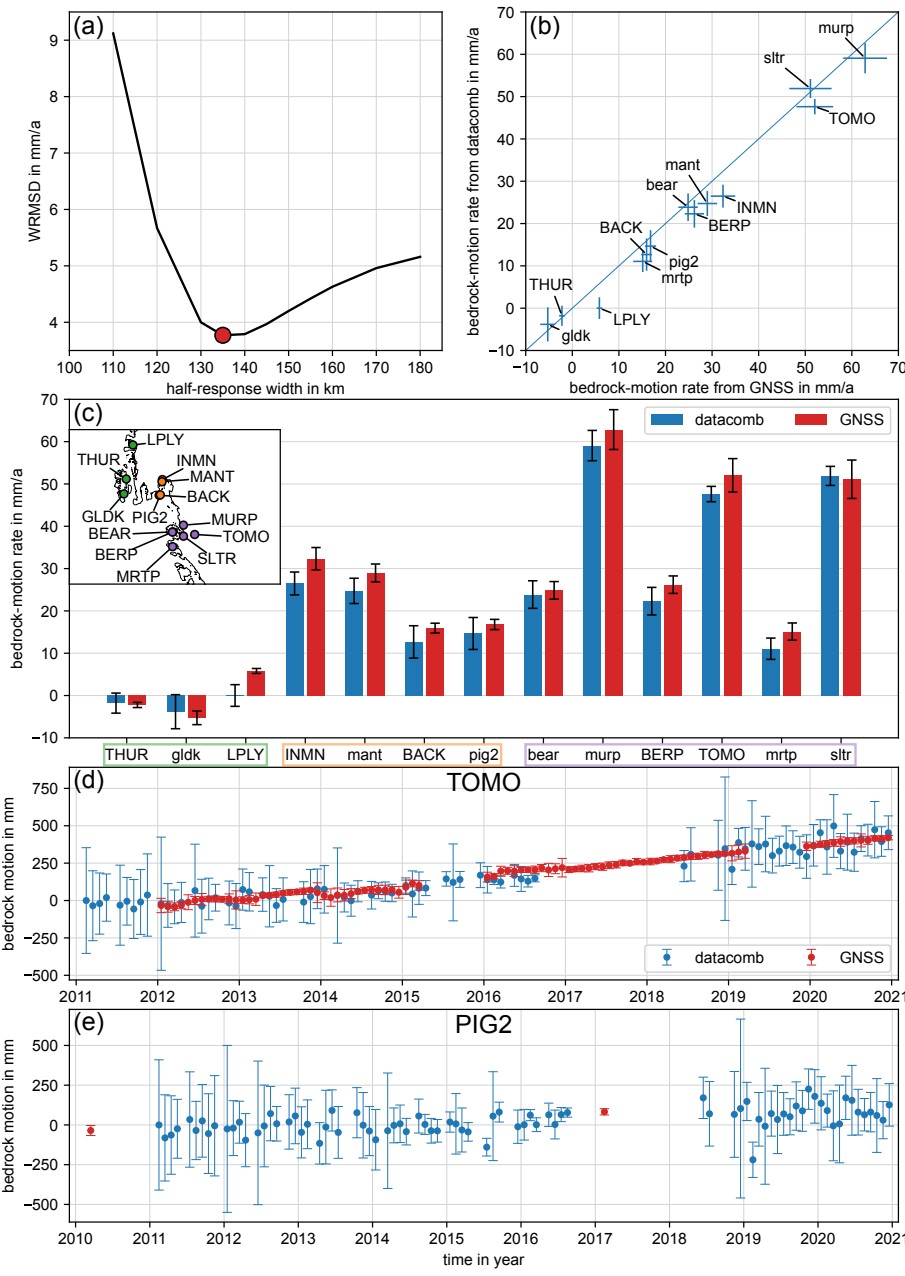

**Figure 2:** Comparison of bedrock motion from data combination (datacomb) and GNSS (GIA + elastic): a) The weighted root mean square difference (WRMSD, Eq. 13) as a function of the Gaussian filter half-response width that we use to filter the input data. The combination results are evaluated at the GNSS sites. Applying a Gaussian filter with 135 km half-response width (red circle) leads to a minimum WRMSD (optimal result). b) A scatter plot of the optimal result showing bedrock motion rates from GNSS vs. data combination. Upper-case site names indicate that more than three years of continuous data is available. c) Bedrock motion rates of the optimal result (blue) compared to GNSS-derived rates (red). The GNSS sites are categorized in three sub-regions: 1 Peripheral Glaciers (green), 2 Pine Island Glacier (orange), 3 Thwaites Glacier (purple). d) and e) illustrating the bedrock motion from the data combination (blue) and GNSS (red) on time series level for the GNSS station TOMO and PIG2 (Table 1), respectively. All uncertainties are 2-$\sigma$.





# 4 Results

We quantify the full bedrock motion due to visco-elastic solid-Earth deformation, i.e. the sum of $h^{\text{GIA}}$ from Eq. 9 and $h^{\text{ELA}}$ derived from the high resolution IMC (Eq. 12, Sect. 3). This enables to compare the result of the data combination directly with the bedrock motion observed with GNSS on bedrock (Eq. 11).

From comparing GNSS data with data combination results, while applying different Gaussian smoothers to the input data, we find the *optimal result* in terms of lowest misfit (Eq. 13), if we choose a half-response width of 135 km (red dot in Figure 2a). For this result, there is high agreement between bedrock motion from the data combination and from GNSS. The results for all stations are very close to the line with a slope of one, which would mean a full agreement (Figure 2b). However, Figure 2b also shows that the results for most

stations are slightly below this line of full agreement. This reveals a small bias of $0.9\,\text{mm a}^{-1}$ (weighted mean of deviations between data combination and GNSS), i.e. on average the absolute magnitudes determined from GNSS are $0.9\,\text{mm a}^{-1}$ larger than from the data combination (Figure 2c). Nevertheless, almost all rates agree within the uncertainties. This is the case for rates with small magnitudes on the peripheral islands (green box in Figure 2c), as well as for high-magnitude rates in the area of the Pine Island Glacier and Thwaites Glacier

(orange and purple box, respectively, in Figure 2c). For the continuously measuring site TOMO, there is a high agreement at the time-series level (Figure 2d). For the episodic site PIG2 (Figure 2e), we find agreement at time-series level, too, however there are only two samples from GNSS. The supplement (Figure S6) contains a comparison on time-series level at all GNSS sites.

Figure 3 illustrates the spatial fields of the mean rates of the determined GIA and IMC of the optimal result.

In the GIA field, we distinguish two local maxima around the Thwaites Glacier and the Pine Island Glacier. Additionally, Figure S1 illustrates the two peaks in a cross section plot. The determined GIA-related bedrock uplift rate peaks at $43 \pm 7\,\text{mm a}^{-1}$ in the Thwaites Glacier region and at $32 \pm 4\,\text{mm a}^{-1}$ in the Pine Island Glacier region. Near these maxima, minima are present at a distance of only $\sim 200\,\text{km}$ (Fig 3a), which are however close to the noise level. This transition from GIA-related uplift to subsidence is also visible in the

GNSS data from Thurston Island (THUR and GLDK in Fig. 2c) and the Pine Island Glacier (INMN, MANT, BACK, and PIG2 in Fig. 2c).

The combination method prescribes that the IMC and FAC results (Figure 3b, Fig. S3) are equally smooth as the resolved GIA. The sign of resolved IMC is opposite to the sign of GIA for large parts. The IMC integrated over the basins 21 and 22 (as defined in Zwally et al. (2012), Fig. 1b) plus a 200 km buffer zone

is $-163 \pm 7\,\text{Gt a}^{-1}$. The apparent GIA mass effect integrated over the same region is $+34 \pm 3\,\text{Gt a}^{-1}$. The integrated FAC change over the same region is $14.6 \pm 6.5\,\text{km}^3\text{a}^{-1}$. We use the 200 km buffer zone to account for signal leakage out of integration area due to smoothing, but we neglect for signal leakage into the integration area as this is expected to be minor (Gunter et al., 2014).





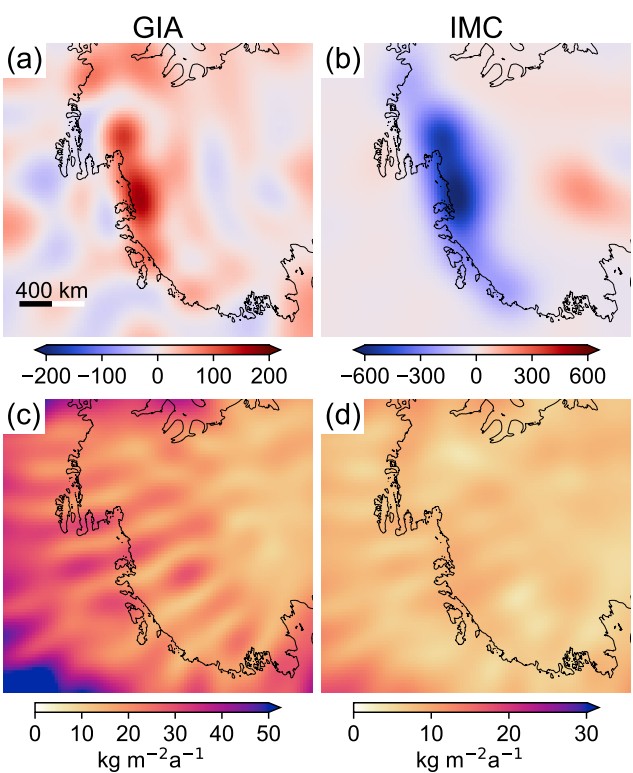

**Figure 3:** Maps of the results for (a) the GIA-induced surface density rate, (b) ice mass change as surface density rate (i.e. that equals a rate of equivalent water height in mm a$^{-1}$). (c) and (d) shows the uncertainties obtained from Willen et al. (2022) propagated to GIA, and IMC, respectively, on trend level. Units are indicated in columns.





# 5  Discussion

## 5.1   Assessment and Comparison

We find that the bedrock-motion mean rates derived from the data combination agree with those observed directly using GNSS within their respective uncertainties (Fig. 2c)). For the first time, we are able to determine present-day GIA effects at such a high spatial resolution from GRACE/GRACE-FO and CryoSat-2 satellite data, thus recovering the vertical bedrock motion independently from GNSS-data in the ASE. Previous similar combination approaches (e.g. Gunter et al., 2014; Engels et al., 2018; Willen et al., 2024) have only been able to determine an overly smoothed estimate of the true GIA effects, which only partly explained the GNSS data. These studies have found GIA-related vertical rates up to about 15 mm/a in the ASE and a spatial pattern with long wavelengths and only one maximum, whereas GNSS observations suggest GIA-related vertical uplift of more than 40 mm/a (Table 1). The restriction to long wavelengths also holds for the inverse approach of Martín-Español et al. (2016b), incorporating GNSS data in addition to gravimetry and altimetry data, as this study constrains the GIA parametrization to length scales of 500 km and larger. Even though the GIA parametrization is prescribed in such a way that shorter spatial scales are allowed in the ASE and the Antarctic Peninsula compared to the remainder of Antarctica, the length scale is still insufficient to resolve the spatial scales that GNSS data suggests. Although Martín-Español et al. (2016b) include GNSS observations in the estimation procedure, there are still large misfits between the GIA estimate and the GNSS data, in particular in the ASE. This remains the case for a modification of this approach in Martín-Español et al. (2016a) that allows for shorter spatial wavelengths in the ASE, but which are presumably still overly large. Likewise, Sasgen et al. (2017) present GIA estimates at a spatial resolution of 200 km, based on an inversion that includes GNSS data, in addition to gravimetry and altimetry data. Even though they resolve GIA-related vertical uplift rates of close to 20 mm/a, still, these magnitudes and spatial scales of estimated GIA are insufficient to explain the GNSS data.

In contrast to previous combination approaches and most GIA forward modelling results (Whitehouse et al., 2019), we can resolve two distinct local maxima of the GIA-related bedrock motion in the ASE: namely in the area of the Pine Island Glacier and in the area of the Thwaites Glacier (Fig. 3a, S4). These two maxima are also postulated by the GIA forward modelling approach of Barletta et al. (2018), which best fits GNSS observations (Fig. 4a+b) when using a low-viscosity upper mantle and a centennial loading history in the ASE. Figure 4 and S5 illustrate the comparison of the GIA vertical motions that we obtain and the modelling result from (Barletta et al., 2018). In the ASE, both approached results provide a similar spatial pattern (smoothness and shape) and magnitude. The difference image (Fig. 4c) shows that the maxima in Pine Island Glacier and Thwaites Glacier largely coincide. Nevertheless, there are significant deviations that can be attributed to limitations of the data combination method (noise in the GIA solution) as well as the modelling approach by Barletta et al. (2018) (strict focus on the ASE).



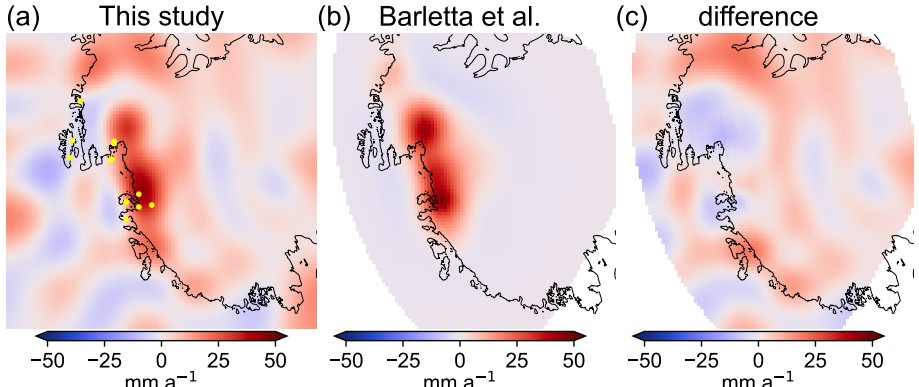

**Figure 4:** (a) The GIA result from the data combination expressed as bedrock motion rate. Yellow dots mark positions of GNSS sites (Fig. 2c). (b) The GIA forward modelling result from (Barletta et al., 2018) with the best fit to the GNSS rates in this region. (c) The difference between (a) and (b).

## 5.2 Spatial resolution

The high spatial resolution of the data combination GIA rates reveals spatial features at a scale less than $\approx$ 200 km, visible in Fig. 3a. Our results indicate a sign reversal on short spatial scales that shows in the GNSS-derived rates too, e.g. by comparing the GIA-related rate at the site THUR of $-4.0 \pm 0.6 \, \text{mm a}^{-1}$ at Thurston
Island and at the site INMN of $23.6 \pm 2.6 \, \text{mm a}^{-1}$ close to Pine Island Glacier (Table 1). We may interpret these as a the resolved forebulge, but, however, some part of these spatial features is likely related to Gibbs artefacts and could only be truly verified by further in situ measurements. A profile of several GNSS sites located approximately along the 100° West meridian from Thurston Island to Pine Island Glacier would be very helpful to investigate this sign reversal of bedrock motion in more detail. This would be, however, also
challenging due to the limited availability of bedrock outcrops.

We attribute our success in resolving the mass change signals at a high spatial resolution to the large signal-to-noise ratio in the ASE: both IMC and GIA signals are large. Moreover, the noise of GRACE/GRACE-FO data is comparatively low locally, as these missions have a polar orbit and the study area is located in a high-latitude region, i.e. the spatial sampling is higher than in areas of lower latitude. In other words, spherical
harmonic coefficients of high degrees and low orders can be determined with more accuracy than coefficients of high degrees and high orders. The half-wavelength of a field given as spherical harmonics is usually used to describe its spatial resolution (e.g. Vishwakarma et al., 2018). A "rule of thumb" is $20000 \, \text{km}/L$ ($L$ maximum degree of development). This might be misleading to assess the spatial resolution of GRACE/GRACE-FO gravity fields on a grid in the spatial domain as the theoretical resolution differs in longitude and latitude
direction due to meridional convergence (e.g. Vishwakarma et al., 2018). Expressed in degrees instead of equatorial circumference, the theoretical resolution is 180°/L in latitude, $\theta$, direction and $\cos(\theta)180°/L$ in longitude direction. For $L = 96$ this results in 1.875° , i.e. the theoretical resolution in latitude direction is



$1.875 \cdot (40000\,\mathrm{km}/360°) \approx 208\,\mathrm{km}$. The latitude of our study region is $\theta = \sim 75°$ South, hence in the longitude direction the resolution is $\cos(\sim 75°) \cdot 1.875° \cdot (40000\,\mathrm{km}/360°) \approx 54\,\mathrm{km}$. This approximation holds for a sphere and is slightly different for an ellipsoid. Moreover, the spatial resolution is affected by the Gaussian smoothing applied on the polar stereographic grid in the spatial domain. The Gaussian smoother equally av-
erages information in longitude and latitude direction. The combined effect may be simply approximated by averaging with $(208\,\mathrm{km} + 54\,\mathrm{km})/2 = 131\,\mathrm{km}$, which is actually close to the Gaussian smoother half-response width of $135\,\mathrm{km}$ that we find for our best fit result.

The evaluation of gravitational fields on an ellipsoid using the method from Ditmar (2018) allows a more realistic assessment of surface density changes. In particular in the ASE region, Ditmar (2018) showed that
the evaluation on a sphere in contrast to an ellipsoid causes up to 15 % difference in the signal magnitude. A test run using a spherical approximation (Wahr et al., 1998) has shown that we need to choose a larger Gaussian filter ($160\,\mathrm{km}$ half-response width) to achieve a best-fit with the GNSS data. From this, we conclude that the ellipsoidal approximation allows for a higher spatial resolution than the spherical approximation in the combination approach, as the ellipsoidal surface allows for a more realistic assignment of the mass change
signal.

## 5.3  Limitations

Our estimation procedure does not take far-field effects and the error covariance structure of GRACE/GRACE-FO into account, as done in some other combination approaches (e.g. Willen et al., 2024), which potentially leads to errors in the result. However, we use the covariance information of the input data sets for estimating
the uncertainties. Willen et al. (2022) indicated that for the grounded AIS, far field effects due to hydrological and glacier mass changes for a 10 year period are on the order of 1 and $10\,\mathrm{Gt\,a^{-1}}$, respectively. If we assume the estimated GIA to fully absorb the far field effects, this would result in a GIA bedrock motion bias of 0.02 and $0.2\,\mathrm{mm\,a^{-1}}$, respectively (assuming an area of the grounded AIS of about $12 \cdot 10^{12}\,\mathrm{m^2}$ and an effective GIA density of $3700\,\mathrm{kgm^{-3}}$ and a uniformly distributed far field effect). We argue that due to the large GIA
signal, errors of less than one millimetre over a 10 year period are hardly relevant, as the signal-to-noise ratio is still large (Fig. 2c). This statement does not apply to other regions of Antarctica and we recommend a thorough error characterization of the input data when determining significant signals outside of the ASE region. This plays a crucial role especially for the evaluation of altimetry over the East Antarctic ice sheet because firn-thickness variations dominate over ice-flow-dynamic changes and assessing firn-thickness variations is a
challenge (Kappelsberger et al., 2023). Furthermore, as mentioned in section 3, there is an inconsistency in the applied methodology when taking into account elastic deformations that occur due to contemporary changes in ice mass (i.e. changes in uplift). We quantify this inconsistency (Sect. S1) and find it is negligible small.

We use the data at monthly resolution, as these are available at this temporal resolution. However, we refrain





from concluding about a temporal variability of the bedrock motion rates based on our results at time-series level (Fig. 2d+e), as the estimates are too noisy and large uncertainties are present. This may also hold for GNSS time series (Koulali and Clarke, 2020). At the "TOMO" station (Fig. 2d), temporal fluctuations are caused in particular by error effects such as accumulated ice in the antennas or equipment changes. Since 2017,

such error effects have been eliminated, e.g. by sealing the antenna. Future studies may investigate whether it is possible to manage these errors and to derive time-variable rates related to transient solid-Earth deformation (Simon et al., 2022). Based on findings from Powell et al. (2020), no dominant magnitudes of viscous effects in the solid earth response are expected with an investigation period of only 10 years and the assumption of a Maxwell rheology of the mantle. Significant effects should be measurable from $\approx$20 years.

The assumptions about the effective densities (Eq. 7+8, Fig. S2) in Eq. 10+9 may also be seen as limitations. We base these assumptions on how to link mass changes to volume changes of the various processes on previous studies (e.g. Riva et al., 2009; Gunter et al., 2014). Investigations with GIA models and ice sheet models may reveal whether further improvements in the estimation can be achieved by refining these effective densities.

## 5.4  Outlook

The estimated present-day GIA with its high spatial resolution is an excellent data set for validating GIA models and coupled GIA-ice-sheet models (Calcar et al., 2023; Albrecht et al., 2024). In contrast to GIA-induced vertical bedrock motion rates from GNSS, the data-combination based result provides information on GIA throughout the ASE region, even where the bedrock is covered with ice. Assumptions on a locally adapted, a 3D or transient rheology in this region, as well as assumptions on the ice loading history may be verified with

the present-day GIA effect determined here.

A monthly temporal resolution is certainly not necessary to determine GIA on time-series level, as we do not expect GIA to fluctuate on these short time scales. Nevertheless, temporal variations of GIA rates due to effects related e.g. to transient rheology are an evolving subject of investigation. GIA modelling based on transient rheology can be helpful to find a temporally reasonable parametrization that ranges between monthly and a

constant rate for future studies.

In other regions of Antarctica, the signal-to-noise ratio is much smaller, so that a more extensive error analysis (e.g. Willen et al., 2024) is required to retrieve sound results. Future satellite gravimetry missions (Daras et al., 2024) and evaluation of time-variable gravity data on trend-level (Loomis et al., 2021; Kvas et al., 2023) might be useful to assess mass changes in the ASE and other regions at a even higher spatial resolution. This may

allow us to further decrease the spatial smoothing applied to the input data sets and could resolve smaller-scale GIA patterns, if they exist. High-quality GNSS data with a favorable spatial coverage would be necessary to validate such investigations.

In this study, we distinguish between an elastic response and GIA as two separate deformation effects. Par-



ticularly in the ASE region, this distinction will not be possible for investigation periods of several decades, as these effects overlap over multi-decadal periods in this region (Powell et al., 2020). In the future, the longer observation time series of 20 years and more will become increasingly relevant to investigate whether we can quantify the visco-elastic deformation effects of the solid Earth that were triggered by the loading changes
during the age of satellite observations and visco-elastic deformation in response to ice loading changes on centennial and millennial time scales.

# 6 Conclusions

This study presents a regional combination method using data from GRACE/GRACE-FO, CryoSat-2, regional climate modelling and firn modelling. For the first time, this combination of data resolves vertical bedrock
motion rates in the Amundsen Sea Embayment that agrees with rates from GNSS on bedrock. We resolved a GIA-induced uplift of more than $40 \, \text{mm} \, \text{a}^{-1}$ at maximum, whereas previous data combination approaches have only resolved less than half of this magnitude at maximum. The results reveal that GIA masks about a quarter of the total observed mass loss in this region from January 2011 to December 2020. We assign $-163 \pm 7 \, \text{Gt} \, \text{a}^{-1}$ to the ice mass change and $+34 \pm 3 \, \text{Gt} \, \text{a}^{-1}$ to the apparent mass effect caused by GIA. Thus,
we determine present-day GIA effects in a region where it is a great challenge to forward model GIA, as both the rheology and the decisive centennial ice-loading history come with significant unknowns (Whitehouse et al., 2019). The large signal-to-noise ratio in this area permits some error contributions to be ignored, so that agreement with GNSS within the errors is still guaranteed. The resulting GIA estimates may be particularly useful for coupled ice-sheet solid-Earth models (Calcar et al., 2023; Albrecht et al., 2024) to study the feedback
between bedrock motion and glacier flow, which may foster improvements of feedback predictions. So far, our approach is justified to resolve long-term (10+ years) temporal variations of bedrock-motion rates only, as short-term variations are dominated by short-term errors of the input data.

# Data availability

GRACE and GRACE-FO monthly gravitational fields can be obtained via https://doi.org/10.5880/ICGEM.2018.003
(Mayer-Gürr et al., 2018).

CryoSat-2 data can be obtained from https://earth.esa.int/eogateway/catalog/cryosat-products.

RACMO2 SMB (Wessem et al., 2018) and IMAU-FDM (Veldhuijsen et al., 2023) are available on reasonable request (https://www.projects.science.uu.nl/iceclimate/).

We will make the optimal result publicly available after review.



## Author Contributions

CRediT: Conceptualization: MW, BW, TB | Data curation: EB, VH, MW | Formal Analysis: MW | Funding acquisition: BW | Investigation: MW, TB | Methodology: MW | Software: MW | Validation: MW, EB | Visualization: MW | Writing – original draft: MW | Writing – review & editing: TB, BW, EB, VH, MW.

## 5 Competing interests

At least one of the (co-)authors is a member of the editorial board of The Cryosphere.

## Acknowledgements

We thank Michiel van den Broeke and his colleagues from the Ice and Climate Group at the Institute for Marine and Atmospheric research Utrecht (IMAU) for providing the regional climate modelling and firn modelling results as well as for the fruitful discussions. We kindly thank all colleagues and institutions who provided geodetic GNSS data in Antarctica to the SCAR-endorsed Geodynamics In ANTarctica based on REprocessing GNSS dAta Initiative (GIANT-REGAIN) led by Mirko Scheinert (TU Dresden, Germany) and Matt King (University of Tasmania, Hobart, Australia). Matthias Willen acknowledges funding from the Nederlandse Organisatie voor Wetenschappelijk Onderzoek (NWO) (project number: C43A13). Taco Broerse acknowledges funding from NWO (grant number ENW.GO.001.005). The work of Eric Buchta was funded by the grant SCHE 1426/26-1 and 2 (project number 404719077) of the Deutsche Forschungsgemeinschaft (DFG) as part of the SPP 1158 "Antarctic Research with Comparative Investigations in Arctic Ice Areas".





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
