# Peer review of "Satellite data reveal details of glacial isostatic adjustment in the Amundsen Sea Embayment, West Antarctica"

_EGUsphere, 2024_

## Author Response (AR1)

We are very pleased about the highly positive assessment of our work. We thank Matt King (RC1) and the anonymous reviewer (RC2) a lot for their helpful comments on our manuscript. Please find below how we revised the manuscript in detail in response to the reviewers' comments. Italic font indicate the referee comments. Green text indicates the authors' responses; in blue we have added text fragments indicating changes in the manuscript. Along with the revised manuscript, we have submitted a complete marked-up version, which highlights all changes made in the revised manuscript.

**Authors' comment to RC1**

*1. Please add a note on the origin of the various frames of GPS and the empirical model.*

We added that the GPS data is consistently in in centre-of-figure reference frame IGb14. The empirical model from the data combination is in the close proximity of a centre-of-figure reference frame, too. During GRACE/GRACE-FO data processing, we add degree-1 coefficients following the approach from Sun et al. (2016). We added to Sect. 2.1, Sect. 2.4, and Sect. 3, respectively:

We thus transfer the gravity fields into a centre-of-figure reference frame.
The GNSS data are in the centre-of-figure reference frame IGb14.
Both, $\dot{h}_{i,\mathrm{GNSS}}^{\mathrm{BM}}$ and $\dot{h}_{i,\mathrm{COMB}}^{\mathrm{BM}}$ are in a centre-of-figure reference frame.

*2. I was also very confused by the text before Equation 2 and Eq2 also. The first part of equation 2 is the daily formal errors. These are well known to be over-optimistic but also entirely dependent on choices in the analysis that are subjective. How were these scaled? The first part of Eq2 sums these in quadrature but there is no multiplier of 1/D. Is that correct? I do not understand the rationale for the two components of Eq2 so a larger explanation would be welcome.*

We calculate monthly time series as the weighted mean of the daily solutions (Eq. 1). As a weight, $w_d$, we use: $w_d = (\sigma_d^{\mathrm{GNSS}})^2$. The uncertainty (variance) of the weighted mean is formally calculated as $1/\Sigma(w_d)$, which we use in the first part of Eq. 2 (e.g. Taylor, 1997, Ch. 7). We agree that it is well known that the daily formal errors are over optimistic. The scatter of the daily solutions within a month is sometimes larger than the formal uncertainties of the daily solution reflect. We argue that the variation of the coordinates within a month represent error effects rather than displacements of the bedrock, i.e. the estimated uncertainty according to $1/\Sigma(w_d)$ is likely to be not representative. We were aiming for a more realistic measure of the uncertainty of the monthly values from the GNSS time series. We decided to pragmatically add to the uncertainty resulting from the formal errors (first part of the equation), the measure of the scatter of the values within a month (second part of the equation). The latter is the variance resulting from the daily values within a month. We argue that this is a more realistic approach than an uncertainty measure that is based on the formal uncertainties only. To clarify this, we extended the explanation of this approach in the text as follows:

 The uncertainty of the monthly weighted averages, $\sigma_m^{\mathrm{GNSS}}$, could be calculated from the formal daily uncertainties (e.g. Taylor, 1997, Ch. 7). However,  we find unphysical scatter of the daily values within a month that is not represented by the formal uncertainties of each daily solution. From this we conclude that the formal uncertainties are likely to be over-optimistic and too limited to represent all  noise sources (Buchta et al., 2024). For this reason, we add  a measure of the scatter of all daily solutions  within a month, i.e. their variance, as an additional measure of uncertainty to the formal uncertainties:

$$(\sigma_m^{\mathrm{GNSS}})^2 = \left[\sum_{d=1}^{D}(\sigma_d^{\mathrm{GNSS}})^{-2}\right]^{-1} + \frac{1}{D-1}\sum_{d=1}^{D}(h_d^{\mathrm{GNSS}} - h_m^{\mathrm{GNSS}})^2 .$$

The first summand represents the uncertainty of the weighted mean derived from the formal daily

uncertainties. The second summand is the variance of the daily expectation values within a month.

*3. P9L5: I was unsure if the spatial mask is exactly the values of Riva et al or something else 'according to' is ambiguous.*

The utilized mask is not exactly the same as used by Riva et al. (2009) but generated very similar. We clarified this in the manuscript as follows:

We use a spatial mask given the GIA density at each location in Antarctica  similar as the one utilized by Riva et al. (2009). They assessed this ratio between GIA-induced gravity changes and GIA-induced geometry changes from GIA forward model outputs following findings from Wahr et al. (2000) and refined it to account for self-gravitation of sea level. Following Riva et al. (2009), we generate the mask by assuming $\rho_{\text{CONTINENT}}^{\text{GIA}} = 4000\frac{\text{kg}}{m^3}$ over the continent and $\rho_{\text{OCEAN}}^{\text{GIA}} = 3400\frac{\text{kg}}{m^3}$ over the ocean. We assume a smooth transition between continent and ocean by using a 100 km Gaussian smoother (Fig. S2). Note that we do not run GIA forward models to tune these densities nor the length of transition between continent and ocean. Riva et al. (2009) found that a $300\,\text{kg}\,\text{m}^{-3}$ increase of the GIA density leads to an 2.5 % increase of the GIA solution.

*4. Section 4: please define the meaning of Gaussian width. there are various definitions used such as half height or 6 sigma or 1 sigma.*

By "Gaussian width", we refer to the "half-response width", i.e. the distance between the centre of the gaussian hat and the value where the maximum has decreased to its half. We clarified this by adding:

It is initially unknown, what Gaussian filter width (here referred to as the half-response width, i.e. the distance between the maximum and its half amplitude) is optimal for separating GIA from IMC such that the spatial resolution is close to the true GIA effect.

*5. Discussion: please come back to the potential origin of the bias with the GNSS of 1mm/yr. Where may this come from? Some informed speculation is appropriate.*

While the bias is within the uncertainties, we agree that the origin of the bias is an interesting question which future work may investigate. We suspect that this bias has been mainly introduced by the input data sets of the combination approach and only partly by the GNSS data. Possible long-term climate trends outside the modelling period of the utilized regional climate model could cause a trend error (Medley and Thomas, 2019) as well as climate trends during the initialization of the firn model (Thomas et al., 2017). Other possible causes, already discussed in Sect. 5.3, are far field effects of mass redistribution within the Earth system that has been not accounted for in the regional combination approach while evaluating the GRACE/GRACE-FO gravity fields. This theoretically includes all other mass changes, including those from the Greenland ice sheet, terrestrial hydrology, glaciers, and GIA outside Antarctica. We expanded the discussion regarding this bias in Sect. 5.3 as follows:

 We find a bias of $0.9\,\text{mm}\,\text{a}^{-1}$ when comparing the GNSS uplift rates with the combination results (Sect. 4). We argue that due to the large GIA uplift of up to $43 \pm 7\,\text{mm}\,\text{a}^{-1}$, errors of less than one millimetre per year over a 10 year period are hardly relevant (Fig. 2c). Systematic errors responsible for the small bias may be: long-term climate trends outside the modelling period of the utilized regional climate model (RACMO2.3p2: 1979–2021), which could cause a trend error (Medley and Thomas, 2019). Furthermore, the firn model needs to be initialized over a reference period to generate an equilibrium firn layer. We assume that there are no dominant climate trends over the reference period, which is 1979–2021 in case of the IMAU-FDM (Veldhuijsen et al., 2023). However, this assumption may not be valid in reality. If there are climate trends during the initialization, this assumption will lead to errors in the trend (Thomas et al., 2017). Even though we argue that the bias is hardly relevant given by

the signal to noise ratio, this may not apply to other regions of Antarctica. We recommend a thorough error characterization of the input data when aiming to determine signals outside of the ASE region. This plays a crucial role especially for the evaluation of altimetry over the East Antarctic ice sheet where firn-thickness variations dominate over ice-flow-dynamic changes and assessing firn-thickness variations remains a challenge (Kappelsberger et al., 2024-09). In addition, there are uncertainties on secular time scales of the GNSS solutions, which are induced by the realization of the Terrestrial Reference Frame. For example, the ITRF2014 shows a drift of $0.2\,\mathrm{mm\,a^{-1}}$ in translation of the Z-coordinate compared to the current realization ITRF2020 (Altamimi et al., 2023), which particularly maps onto the vertical velocities in polar regions.

Moreover, our estimation procedure does not take far-field effects and the error covariance structure of GRACE/GRACE-FO into account, as done in some other combination approaches (e.g. Willen et al., 2024), which potentially leads to errors in the result. However, we use the covariance information of the input data sets for estimating the uncertainties introduced by far-field signals. Willen et al. (2022) indicated that for the grounded AIS,  far-field effects due to hydrological and glacier mass changes for a 10 year period are on the order of 1 and $10\,\mathrm{Gt\,a^{-1}}$, respectively. If we assume the estimated GIA to fully absorb the far field effects, this would result in a GIA bedrock motion bias of 0.02 and $0.2\,\mathrm{mm\,a^{-1}}$, respectively (assuming an area of the grounded AIS of about $12{\cdot}10^{12}\,\mathrm{m^2}$ and an effective GIA density of $3700\,\mathrm{kg\,m^{-3}}$ and a uniformly distributed far field effect).

The assumptions about the effective densities (Eq. 7+8, Fig. ~~2c) . This statement does not apply to other regions of Antarctica and we recommend a thorough error characterization of the input data when determining significant signals outside of the ASE region. This plays a crucial role especially for the evaluation of altimetry over the East Antarctic ice sheet because firn-thickness variations dominate over ice-flow-dynamic changes and assessing firn-thickness variations is a challenge . Furthermore, as~~ S2) in Eq. 9+10 may also contribute to the bias. We base the relation between mass changes and volume changes of the various processes on previous studies (e.g. Riva et al., 2009; Gunter et al., 2014). Investigations with GIA models and ice sheet models may reveal whether further improvements in the estimation can be achieved by refining these effective densities.

*6. Please also discuss a little more what GIA modelling went into the Riva mask and if that explored low viscosity mantle and if not, what impact that could have.*

Regarding the GIA density mask, please refer to our comment above how we revised the manuscript to address this. So far, we have not yet carried out any investigations with GIA models to clarify whether a lower viscosity in the upper mantle impacts the relationship between GIA-induced gravity changes and GIA-induced geometry changes, i.e. leads to a higher or lower "GIA density". In our study, we assume that the GIA density does not depend on the rheology, as the GIA density is a ratio of effects and not a volumetric mass density in terms of a material property. However, this is an assumption and could be clarified by studying (3-D) GIA modelling outputs in future work. Riva et al. (2009) wrote that an $300\,\mathrm{kg\,m^{-3}}$ increase of the GIA density leads to an 2.5 % increase of the GIA signal. This could indicate that the assumed density is not a major source of uncertainty.

*7. General: I think spatial high resolution is not the right term but spatially continuous. Consider changing throughout.*

We agree and changed throughout.

*8. P2L5 add the Gomez et al 2024 study (Science Advances)*

Done.

*9. P6L5-7 I did not understand what was meant here.*

This was a complicated description for surface melting, e.g. in blue ice areas. In this case, there would be an SMB-induced mass change that is not part of the firn layer. We removed this sentence as we already state that we assume that the mass change of the firn layer equals the SMB change.

*10. P6 L23 "At some" -> "Some"*

Done.

*11. P13L21 should be Fig S4 I think*

The two local maxima can be seen in Fig. 3a and S4. We reformulate the sentence as follows:

Figure 3 illustrates  maps of the mean rates of the determined GIA and IMC  derived from the optimal result  (cf. Fig. S4 for a cross section across the region highlighting the dominant signals).

*12. P13L39 please repeat the data period given the trend is limited to that for IMC*

We included 2011-01 to 2020-12.

*13. P5L6 well, it is tuned to GNSS in terms of filter width*

We agree that "independently" might be misleading and reformulate the sentence as follows:

 Our spatially-continuous inverse GIA estimate is the first to agree with vertical bedrock velocities from GNSS data in the ASE.

*14. P15L8 this sentence is only for ASE. I note Wolstencroft et al found good agreement in the southern peninsula, updated by Koulali et al 2023*

We agree and added ASE to make clear that this sentence does not hold for other regions.

*15. L16L6 'as a the'*

Done.

*16. P16L10 add bedrock locations to Fig4c? and 100W meridian?*

Thanks for this suggestion. We added both to Fig. 4c and extented the figure caption as follows:

The grey box marks the region of sign reversal of bedrock motion that may be of interest for further investigation. The green line shows the 100° West Meridian and the olive dots are rock outcrops from Burton-Johnson et al. (2016) provided via Quantarctica3 (Matsuoka et al., 2021).

*17. P18L7 not sure what 'no dominant magnitudes of viscous effects' means*

We reformulate this as follows:

Based on findings from Powell et al. (2020),  we do not expect that significant rate changes related to viscous

deformation caused by recent IMC are detectable over an investigation period of only 10 years. When assuming a low upper-mantle viscosity, significant viscous effects should be measurable from ≈20 years onwards.

*18. Data: need link to GNSS*

We included the link.

**Authors' comment to RC2**

*1. Throughout: where the text mentions "GIA result" or "estimated present-day GIA" e.g. p18,L15, it would be better to clarify which part of GIA you're talking about. GIA uplift, or GIA-related mass changes etc. There are several places where this clarification would be beneficial.*

We agree and changed throughout at most places. We think that bedrock motion, i.e. uplift, will be most interesting for GIA modelling. In places where the statement applies to both the geometric effect and the gravity effect, we have left GIA result as an umbrella term.

*2. Pg1L12: "GIA result" – do you mean bedrock uplift?*

We reformulate this to:

GIA-related bedrock motion

*3. Pg2L5: is Groh et al. (2012) the right reference here?*

To our knowledge, this is the first peer-reviewed reference reporting the high uplift rates measured with GNSS in the ASE. For example, Thomas et al. (2011) did an Antarctic wide comparison of GNSS data with GIA models but had still not included any data in the ASE.

*4. Pg2L16: "Global GIA models" change to "Global 1D GIA models"*

Done.

*5. Figure 1c: It would be useful to be able to see on this map which GNSS are continuous, and which are campaign. I suggest using a different symbol for each.*

We updated the figure. Sites which are campaign sites only are indicated now with an asterisk beside the name. Unfortunately the symbols overlap. We added the following to the caption of Fig. 1

The asterisk '*' indicates sites that have been observed episodically only.

*6. Pg7L8: "GIA effects" what specific effects – GIA related mass change? Uplift?*

We aim to quantify both and reformulate this as follows:

In order to quantify  GIA-related bedrock motion and GIA-related gravity changes in the ASE,

*7. Pg9L9: "based on findings from GIA modelling" What GIA modelling? This could do with some further explanation as Figure S2 also does not clarify. Also, previous line – "GIA at each location" do you mean the 20x20km grid, or what locations?*

We agree that this was not a sufficient explanation. Please also refer to our comment to the other referee. We now clarified this as follows:

We use a spatial mask given the GIA density at each location in Antarctica  similar as the one utilized by Riva et al. (2009). They assessed this ratio between GIA-induced gravity changes and GIA-induced geometry changes from GIA forward model outputs following findings from Wahr et al. (2000) and refined it to account for self-gravitation of sea level. Following Riva et al. (2009), we generate the mask by assuming $\rho_{\mathrm{CONTINENT}}^{\mathrm{GIA}} = 4000 \frac{\mathrm{kg}}{m^3}$ over

the continent and $\rho_{\text{OCEAN}}^{\text{GIA}} = 3400\frac{\text{kg}}{m^3}$ over the ocean. We assume a smooth transition between continent and ocean by using a 100 km Gaussian smoother (Fig. S2). Note that we do not run GIA forward models to tune these densities nor the length of transition between continent and ocean. Riva et al. (2009) found that a $300\,\text{kg}\,\text{m}^{-3}$ increase of the GIA density leads to an $2.5\,\%$ increase of the GIA solution.

*8. Pg15L28: how low is the upper mantle viscosity used in that study?*

We added $4 \cdot 10^{18}\,\text{Pa}\,\text{s}$.

*9. Pg15L29 – correct reference format*

Done.

*10. Pg16L2: "GIA rates" > "GIA uplift rates"*

Done.

*11. Pg16L6: remove "but" or "however"*

We removed "however".

*12. Pg18L7-10: not sure I understand– "no dominant magnitudes" – also I would expect to see viscous effects on a 10-year time scale in the ASE due to low viscosity.*

We agree that this was misleading and was also criticized by the other referee. Powell et al. (2020) investigated viscous effects in solid-Earth deformation towards present-day ice mass changes and how they would be reflected in GNSS results in the ASE. Among other things, they compared vertical bedrock motion due to visco-elastic deformation under the assumption of a 3D rheology model with the simplification that bedrock motion due to recent ice mass changes induces elastic deformation only. Figure 6 in Powell et al. (2020) shows that from about 15 to 20 years of observation time with GNSS, the error is already several millimetres if the deformation due to recent ice mass changes is assumed to be only elastic. We have now clarified this, also in response to RC1, as follows:

Based on findings from Powell et al. (2020),  we do not expect that significant rate changes related to viscous deformation caused by recent IMC are detectable over an investigation period of only 10 years. When assuming a low upper-mantle viscosity, significant viscous effects should be measurable from $\approx$20 years onwards.

**References**

Altamimi, Z., P. Rebischung, X. Collilieux, L. Métivier, and K. Chanard (2023). "ITRF2020: an augmented reference frame refining the modeling of nonlinear station motions". In: *Journal of Geodesy* 97.5, p. 47. ISSN: 1432-1394. DOI: 10.1007/s00190-023-01738-w.

Buchta, E., M. Scheinert, M. A. King, T. Wilson, A. Koulali, P. J. Clarke, D. Gómez, E. Kendrick, C. Knöfel, and P. Busch (2024). "Advancing geodynamic research in Antarctica: Reprocessing GNSS data to infer consistent coordinate time series (GIANT-REGAIN)". In: *Earth System Science Data Discussions* 2024, pp. 1–31. DOI: 10.5194/essd-2024-355.

Burton-Johnson, A., M. Black, P.T. Fretwell, and J. Kaluza-Gilbert (2016). "An automated methodology for differentiating rock from snow, clouds and sea in Antarctica from Landsat 8 imagery: a new rock outcrop map and area estimation for the entire Antarctic continent". In: *The Cryosphere* 10.4, pp. 1665–1677. DOI: 10.5194/tc-10-1665-2016.

Groh, A., H. Ewert, M. Scheinert, M. Fritsche, A. Rülke, A. Richter, R. Rosenau, and R. Dietrich (2012). "An investigation of Glacial Isostatic Adjustment over the Amundsen Sea sector, West Antarctica". In: *Global and Planetary Change* 98-99, pp. 45–53. DOI: 10.1016/j.gloplacha.2012.08.001.

Gunter, B.C., O. Didova, R.E.M. Riva, S.R.M. Ligtenberg, J.T.M. Lenaerts, M.A. King, M.R. van den Broeke, and T. Urban (2014). "Empirical estimation of present-day Antarctic glacial isostatic adjustment and ice mass change". In: *The Cryosphere* 8.2, pp. 743–760. DOI: 10.5194/tc-8-743-2014.

Kappelsberger, M.T., M. Horwath, E. Buchta, M.O. Willen, L. Schröder, S.B.M. Veldhuijsen, P. Kuipers Munneke, and M.R. van den Broeke (2024-09). "How well can satellite altimetry and firn models resolve Antarctic firn thickness variations?" In: *The Cryosphere* 18.9, pp. 4355–4378. ISSN: 1994-0424. DOI: 10.5194/tc-18-4355-2024.

Matsuoka, K., A. Skoglund, G. Roth, J. de Pomereu, H. Griffiths, R. Headland, B. Herried, K. Katsumata, A. Le Brocq, K. Licht, F. Morgan, P.D. Neff, C. Ritz, M. Scheinert, T. Tamura, A. Van de Putte, M. van den Broeke, A. von Deschwanden, C. Deschamps-Berger, B. Van Liefferinge, S. Tronstad, and Y. Melvær (2021). "Quantarctica, an integrated mapping environment for Antarctica, the Southern Ocean, and sub-Antarctic islands". In: *Environmental Modelling & Software* 140, p. 105015. ISSN: 1364-8152. DOI: https://doi.org/10.1016/j.envsoft.2021.105015.

Medley, B. and E.R. Thomas (2019). "Increased snowfall over the Antarctic Ice Sheet mitigated twentieth-century sea-level rise". In: *Nature Climate Change* 9. ISSN: 1758-678X. DOI: 10.1038/s41558-018-0356-x.

Powell, E., N. Gomez, C. Hay, K. Latychev, and J.X. Mitrovica (2020). "Viscous Effects in the Solid Earth Response to Modern Antarctic Ice Mass Flux: Implications for Geodetic Studies of WAIS Stability in a Warming World". In: *Journal of Climate* 33.2, pp. 443–459. ISSN: 1520-0442. DOI: 10.1175/jcli-d-19-0479.1.

Riva, R.E.M., B.C. Gunter, T.J. Urban, Bert L.A. Vermeersen, R.C. Lindenbergh, M.M. Helsen, J.L. Bamber, R.S.W. van de Wal, M.R. van den Broeke, and B.E. Schutz (2009). "Glacial Isostatic Adjustment over Antarctica from combined ICESat and GRACE satellite data". In: *Earth and Planetary Science Letters* 288.3-4, pp. 516–523. DOI: 10.1016/j.epsl.2009.10.013.

Sun, Y., R.E.M. Riva, and P. Ditmar (2016). "Optimizing estimates of annual variations and trends in geocenter motion and $J_2$ from a combination of GRACE data and geophysical models". In: *J. Geophys. Res. Solid Earth* 121.11, pp. 8352–8370. ISSN: 2169-9313. DOI: 10.1002/2016JB013073.

Taylor, J. R. (1997). *An introduction to error analysis : the study of uncertainties in physical measurements*. 2nd ed. University Science Books, Sausalito, California. ISBN: 0-935702-42-3.

Thomas, E.R., J.M. van Wessem, J. Roberts, E. Isaksson, E. Schlosser, T.J. Fudge, P. Vallelonga, B. Medley, J. Lenaerts, N. Bertler, M.R. van den Broeke, D.A. Dixon, M. Frezzotti, B. Stenni, M. Curran, and A.A. Ekaykin (2017). "Regional Antarctic snow accumulation over the past 1000 years". In: *Climate of the Past* 13. ISSN: 1814-9332. DOI: 10.5194/cp-13-1491-2017.

Thomas, I.D., M.A. King, M.J. Bentley, P.L. Whitehouse, N.T. Penna, S.D.P. Williams, R.E.M. Riva, D.A. Lavallee, P.J. Clarke, E.C. King, R.C.A. Hindmarsh, and H. Koivula (2011). "Widespread low rates of Antarctic glacial isostatic adjustment revealed by GPS observations". In: *Geophysical Research Letters* 38. ISSN: 0094-8276. DOI: 10.1029/2011GL049277.

Veldhuijsen, S.B.M., W.J. van de Berg, M. Brils, P. Kuipers Munneke, and M.R. van den Broeke (2023). "Characteristics of the 1979–2020 Antarctic firn layer simulated with IMAU-FDM v1.2A". In: *The Cryosphere* 17.4, pp. 1675–1696. ISSN: 1994-0424. DOI: 10.5194/tc-17-1675-2023.

Wahr, J., D. Wingham, and C. Bentley (2000). "A method of combining ICESat and GRACE satellite data to constrain Antarctic mass balance". In: *Journal of Geophysical Research* 105. DOI: 10.1029/2000JB900113.

Willen, M.O., M. Horwath, E. Buchta, M. Scheinert, V. Helm, B. Uebbing, and J. Kusche (2024). "Globally consistent estimates of high-resolution Antarctic ice mass balance and spatially resolved glacial isostatic adjustment". In: *The Cryosphere* 18.2, pp. 775–790. DOI: `10.5194/tc-18-775-2024`.

Willen, M.O., M. Horwath, A. Groh, V. Helm, B. Uebbing, and J. Kusche (2022). "Feasibility of a global inversion for spatially resolved glacial isostatic adjustment and ice sheet mass changes proven in simulation experiments". In: *J. Geod.* 96.10, pp. 1–21. ISSN: 0949-7714. DOI: `10.1007/s00190-022-01651-8`.